# Molecular Mechanisms of Herbicide Resistance in Rapeseed: Current Status and Future Prospects for Resistant Germplasm Development

**DOI:** 10.3390/ijms26178292

**Published:** 2025-08-26

**Authors:** Decai Liu, Shicheng Yu, Biaojun Ji, Qi Peng, Jianqin Gao, Jiefu Zhang, Yue Guo, Maolong Hu

**Affiliations:** 1Institute of Industrial Crops, Jiangsu Academy of Agricultural Sciences, Nanjing Sub-Center, National Center of Oil Crops Improvement, Key Laboratory of Cotton and Rapeseed (Nanjing), Ministry of Agriculture, Nanjing 210014, China; decailiu91@163.com (D.L.); y1811769829@163.com (S.Y.); pengqi1981@jaas.ac.cn (Q.P.); chinagjq@163.com (J.G.); jiefu_z@163.com (J.Z.); 2Key Laboratory of Genetics, Breeding and Multiple Utilization of Crops, Ministry of Education/Key Laboratory of Biological Breeding for Fujian and Taiwan Crops, Ministry of Agriculture and Rural Affairs/College of Agriculture, Fujian Agriculture and Forestry University, Fuzhou 350002, China; billjj@126.com; 3School of Food and Biological Engineering, Jiangsu University, Zhenjiang 212013, China; 4Institute of Life Sciences, Jiangsu University, Zhenjiang 212013, China

**Keywords:** germplasm development, herbicide resistance, molecular mechanism, rapeseed

## Abstract

Rapeseed (*Brassica napus*) is a globally important oilseed crop whose yield and quality are frequently limited by weed competition. In recent years, there have been significant advances in our understanding of herbicide-resistance mechanisms in rapeseed and in the development of herbicide-resistant rapeseed germplasm. Here, we summarize the molecular mechanisms of resistance to three herbicides: glyphosate, glufosinate, and acetolactate synthase (ALS) inhibitors. We discuss progress in the identification of new resistance genes and the development of herbicide-resistant rapeseed germplasm, from the initial identification of natural mutants to artificial mutagenesis screening, introduction of exogenous resistance genes, and gene editing. In addition, we describe how synthetic biology and directed protein evolution will contribute to precision-breeding efforts in the near future. This is the first review to systematically integrate non-target resistance mechanisms and the potential applications of multi-omics and AI technologies for breeding of herbicide-resistant rapeseed, together with strategies for managing the risks associated with gene flow, the evolution of herbicide-resistant weeds, and the occurrence of volunteer plants resulting from deployment of herbicide-resistant rapeseed. By synthesizing current knowledge and future trends, this review provides guidance for safe, effective, and innovative approaches to the sustainable development of herbicide-resistant rapeseed.

## 1. Introduction

Rapeseed (*Brassica napus*) is one of the world’s four major oil crops and plays a crucial role in global agriculture [1,2]. According to the Food and Agriculture Organization (FAO), global rapeseed production reached 87.5 million tons in 2022, accounting for 10.6% of total oil crop production. Rapeseed cultivation is threatened by various biotic stresses, among which weed infestation is a primary constraint to both yield and quality [3].

Weeds compete with rapeseed for essential resources, such as light, water, nutrients, and space throughout their growth cycle, and the presence of weeds can create environments conducive to the proliferation of pests and diseases, further compromising yield and quality [4]. Weeds are estimated to reduce annual rapeseed yields by 23–64% compared with yields under weed-free conditions [5].

Traditional weed control in rapeseed fields relies primarily on tillage and manual removal, making it both labor-intensive and resource-consuming. As urbanization accelerates, rural labor shortages in major rapeseed-producing regions have become increasingly significant. Consequently, chemical application has become the predominant weed-control method because of its high efficiency. Despite their effectiveness, conventional herbicides often lack selectivity, causing phytotoxic effects on rapeseed and leading to significant yield losses, particularly with herbicides that target broadleaf weeds. Therefore, achieving efficient weed control while ensuring crop safety has become a major bottleneck in the chemical management of rapeseed fields. Two main strategies have been proposed to address this challenge: the development of new selective herbicides and the breeding of herbicide-resistant rapeseed varieties. The latter approach offers considerable economic advantages. The coordinated deployment of herbicide-resistant varieties together with compatible herbicides can establish a robust “crop safety–efficient weed control” system, making the breeding of herbicide-resistant rapeseed a critical research focus [6,7].

In recent decades, advances in crop-resistance research and breeding technologies have transformed the development of herbicide-resistant rapeseed germplasm [8]. Strategies have progressed from the initial identification of natural mutants to artificial mutagenesis screening, introduction of exogenous resistance genes, and, most recently, use of gene editing technologies for targeted modification. These diverse approaches have enabled the successful development of various herbicide-resistant rapeseed resources. Transgenic technology has enabled the development of rapeseed germplasms resistant to glyphosate, glufosinate, and bromoxynil, and non-transgenic methods have produced germplasms resistant to triazines and acetolactate synthase (ALS) inhibitors. Globally, the herbicide-resistant rapeseed varieties that have been commercially cultivated are primarily those resistant to glyphosate, glufosinate, and ALS inhibitors.

Current research on herbicide resistance in rapeseed is primarily focused on target resistance mechanisms; non-target resistance mechanisms have received less attention. This has restricted the development of rapeseed germplasm with multi-herbicide resistance. Traditional breeding techniques, such as screening for natural or artificial mutations, are hindered by inherent randomness, lengthy genetic improvement cycles, and high workloads. Transgenic technology has raised concerns about biosafety and food safety owing to the introduction of exogenous genes. Although CRISPR/Cas9 gene editing can effectively address the limitations of traditional breeding, the risk of off-target effects has not been fully eliminated. These practical challenges have prompted researchers to focus on the promise of novel breeding techniques for the creation of herbicide-resistant rapeseed germplasm.

Here, we synthesize recent research on the mechanisms of action and plant resistance associated with glyphosate, glufosinate, and ALS inhibitors. We summarize recent advances in the development of herbicide-resistant rapeseed germplasm and discuss future research directions to facilitate the breeding of new resistant varieties. Finally, we address potential challenges associated with the use of herbicide-resistant rapeseed, including gene flow, the evolution of herbicide-resistant weeds and herbicide-resistant volunteer rapeseeds. The information summarized here provides a theoretical framework to guide the safe research, development, and industrial deployment of herbicide-resistant rapeseed.

## 2. Modes of Action of Three Major Herbicides and the Molecular Mechanisms of Crop Resistance

There are numerous important metabolic processes during plant growth. Herbicides primarily function by interfering with and inhibiting one or more key metabolic processes in plants, thereby causing the death of target plants. ALS inhibitors, glyphosate, and glufosinate all achieve their herbicidal effects by inhibiting amino acid synthesis or ammonia metabolism. However, their target sites differ from one another.

### 2.1. ALS Inhibitor Herbicides and ALS

ALS is a flavoprotein encoded by nuclear genes and localized in chloroplasts. It serves as the rate-limiting enzyme in the branched-chain amino acid biosynthesis pathway in plants (Figure 1). ALS facilitates the condensation of two pyruvate molecules to form acetolactate, a precursor for valine and leucine synthesis [9]. Alternatively, it catalyzes the reaction between pyruvate and 2-ketobutyrate to produce 2-acetyl-2-hydroxybutyrate, a precursor for isoleucine synthesis [10]. ALS-inhibiting herbicides (ALS inhibitors) are a class of selective herbicides that target ALS by binding to its active sites, thereby blocking substrate binding and catalytic activity. This process occurs in the cytoplasmic matrix, and this inhibition leads to a deficiency in branched-chain amino acid [11,12], which in turn impairs protein synthesis and cell division, ultimately causing the death of susceptible plants [13,14,15]. The resistance mechanism of ALS herbicides occurs in the cytoplasmic matrix. On the basis of their structural differences, ALS inhibitors are categorized into five main chemical classes: sulfonylureas (SUs), imidazolinones (IMIs), sulfonylaminocarbonyl triazolinones (SCTs), pyrimidinylthiobenzoates (PTBs), and triazolopyrimidines (TPs) [16,17]. ALS inhibitors account for 15–20% of the global herbicide market and are among the most widely used herbicides worldwide. Notably, because ALS is present only in plants and microorganisms, these herbicides are considered non-toxic to animals and are recognized for their high ecological safety [18].

The structure of ALS is crucial for understanding plant herbicide resistance. Crystal structure analyses have revealed that ALS inhibitor herbicides bind to the active site channel of the ALS catalytic subunit, thereby exerting their inhibitory effects. However, the specific binding modes and the induced structural changes differ among herbicide classes. In resistant weeds, common single-point mutations often occur at key positions within the herbicide-binding channel, reducing herbicide affinity through steric hindrance or disrupting the hydrogen bond network. For example, the P_197_L mutation alters the α-helix conformation, hindering the entry of sulfonylureas (SUs) into the binding channel, and the W_574_L mutation causes resistance to multiple herbicide classes by disrupting π-stacking interactions [19,20,21]. Herbicide binding can also induce modifications to the thiamine diphosphate (ThDP) cofactor, and resistance mutations may affect the stability or modification efficiency of the cofactor, indirectly enhancing resistance [22,23]. The regulatory subunit of ALS can also contribute to resistance by influencing the conformational dynamics of the catalytic subunit. Mutations in the small subunit of *Escherichia coli* ALS have been shown to reduce sensitivity to feedback inhibition and to simultaneously enhance herbicide tolerance [24]. These structure–function relationships provide a molecular basis for understanding the mechanisms of herbicide resistance and offer potential target sites for directed evolution of ALS proteins in future engineering efforts.

### 2.2. Glyphosate and EPSPS

Glyphosate targets 5-enolpyruvylshikimate-3-phosphate synthase (EPSPS) [25], an enzyme that catalyzes the reaction between phosphoenolpyruvate (PEP) and shikimate-3-phosphate (S3P) in the shikimate pathway to produce 5-enolpyruvylshikimate-3-phosphate (EPSP). Chorismate synthase subsequently converts EPSP to chorismate, a precursor for the synthesis of aromatic amino acids, including tryptophan, tyrosine, and phenylalanine, as well as secondary metabolites such as lignin and plant hormones. Glyphosate, a structural analog of PEP, competes with PEP in chloroplasts to form a stable glyphosate–S3P–EPSPS complex that irreversibly inhibits EPSPS activity and blocks EPSP biosynthesis (Figure 2). This disruption impedes the production of essential aromatic amino acids and causes excessive accumulation of shikimate. Consequently, the synthesis of hormones and secondary metabolites necessary for plant growth is inhibited, resulting in metabolic disorders and ultimately plant death [26,27,28,29].

On the basis of sequence characteristics and glyphosate sensitivity, EPSPS enzymes are categorized into Class I (sensitive) and Class II (resistant). Class I EPSPSs are widely distributed in plants and Gram-negative bacteria and are strongly inhibited by low glyphosate concentrations. By contrast, Class II EPSPSs, which have been isolated from some microorganisms (e.g., *Agrobacterium* sp. CP4, *Achromobacter* sp. LBAA), exhibit inherently high resistance to glyphosate [30]. In the Class II enzyme CP4-EPSPS, conserved glycine and proline residues are substituted by an alanine at position 100 and a leucine at position 105, respectively. These substitutions enable CP4-EPSPS to maintain normal binding with its substrate PEP while adopting a conformation that precludes glyphosate binding [31]. Leveraging this characteristic, Monsanto developed glyphosate-resistant corn by overexpressing the *CP4-EPSPS* gene. Class I EPSPS can also maintain catalytic function while reducing glyphosate affinity through specific mutations. A prime example is the Thr_179_Ile and Pro_183_Ser double mutation in *Capsicum annuum* EPSPS [32]. Structural analyses have shown that Thr_179_Ile shifts Gly_178_, thereby weakening glyphosate binding, while Pro_183_Ser alters the enzyme’s conformation to improve PEP utilization. The Thr_179_Ile mutation alone reduces PEP affinity, making the compensatory Pro_183_Ser mutation essential for stable resistance [33]. These findings reveal the molecular basis of glyphosate resistance in plants.

### 2.3. Glufosinate and GS

Glufosinate is a non-selective, phosphoric-acid-based herbicide whose active ingredient, L-phosphinothricin (L-PPT), is a derivative of secondary metabolites produced by *Streptomyces* spp. and has a chemical structure similar to that of glutamic acid. Glufosinate targets glutamine synthetase (GS), an enzyme essential for the synthesis of glutamine from glutamic acid and NH_4_^+^, thereby playing a crucial role in plant nitrogen metabolism and intracellular nitrogen balance. By competitively inhibiting GS [15], glufosinate prevents plants from assimilating ammonium ions generated by photorespiration and nitrogen metabolism into glutamine, causing a substantial increase in intracellular ammonium levels (Figure 3). Increased ammonium levels compromise cell-membrane integrity, alter pH balance, and hinder photosynthetic light reactions, notably by damaging the thylakoid membrane. This accumulation of ammonium indirectly promotes the production of reactive oxygen species (ROS), like superoxide anions and hydrogen peroxide, which induce oxidative stress and damage cellular components, including lipids, proteins, and DNA, thereby expediting cell death [34,35]. This mechanism occurs in the chloroplasts of plants.

The evolution of glufosinate resistance in weeds has been relatively slow, with resistance reported in only one goose grass and three ryegrass species to date. At the molecular level, studies of Italian ryegrass (*Lolium multiflorum*) have identified a substitution mutation at position 171 of the GS protein, in which aspartic acid is replaced by asparagine, leading to a marked reduction in the enzyme’s glufosinate sensitivity [36]. In alfalfa, resistance is observed when glycine occupies position 207, a non-glycine residue is present at position 245, and arginine or lysine is found at position 332 [37]. In rice, a mutant form of glutamine synthetase (OsGS1;1) enhances glufosinate tolerance through the combined effects of glycine at position 59 and arginine at position 296 [38]. These studies have provided initial insight into the molecular mechanisms of plant resistance to glufosinate at the level of protein structure and function.

## 3. Research Progress in the Development of Herbicide-Resistant Rapeseed Germplasm

Over recent decades, advancements in crop resistance research and breeding technologies have significantly transformed the development of herbicide-resistant rapeseed germplasm. The strategies employed have progressed from the initial identification of natural mutants to artificial mutagenesis screening, the introduction of exogenous resistance genes, and, most recently, the application of gene editing technologies for targeted modification. These diverse approaches have led to the successful development of various herbicide-resistant rapeseed resources.

### 3.1. Development of Herbicide-Resistant Rapeseed by Natural Mutation

During natural evolution, nucleotide mutations occur at a rate of approximately 10^−9^ per base pair. Numerous studies have established a link between such spontaneous mutations and the emergence of herbicide resistance in weeds. For instance, mutations in the *ALS* gene of *Descurainia sophia* (substitutions of Asp-376 with Glu and Trp-574 with Leu) increase tribenuron-methyl (TBM) resistance 815-fold and 366-fold, respectively [39]. In addition, mutations of Pro-197 to Leu, His, Ser, or Thr confer a 150- to 211-fold increase in resistance. Investigations of the *EPSPS* gene have revealed that simultaneous mutations at Thr-102 and Pro-106 in *Eleusine indica* and *Bidens subalternans* increase glyphosate resistance 15- to 31-fold [40], and mutations at Thr-102, Ala-103, and Pro-106 in *EPSPS* of *Amaranthus hybridus* produce a 314-fold increase in glyphosate resistance [41]. Studies on herbicide-resistant rapeseed germplasm also underscore the significance of natural mutations. The earliest resistant germplasm was derived from naturally mutated plants. Subsequent screening efforts at the field, seedling, or protoplast level have led to the identification of numerous herbicide-resistant variants, most of which possess mutations in the *ALS* gene and exhibit resistance to ALS inhibitor herbicides.

The genome of *Brassica napus* (AACC, 2n = 38) contains five *ALS* homologs (*BnALS1*–*BnALS5*): *BnALS2*, *BnALS3*, and *BnALS4* on the A genome and *BnALS1* and *BnALS5* on the C genome [42,43]. *BnALS1* and *BnALS3* are constitutively expressed and share over 98% amino acid sequence identity. By contrast, *BnALS2* is predominantly expressed in flower buds, flowers, and siliques and shows approximately 75% similarity with *BnALS1*/*BnALS3*. *BnALS4* and *BnALS5* have been characterized as pseudogenes [44]. Resistance to ALS inhibitors in rapeseed is primarily conferred by mutations in the *BnALS1* and *BnALS3* loci. The earliest rapeseed germplasm resistant to ALS inhibitors originated from an SU-herbicide-resistant mutant identified by Miki from *Brassica napus* protoplast culture, in which the resistance phenotype showed single-gene dominant inheritance [45]. In 2004, the naturally occurring imidazolinone (IMI)-resistant rapeseed line M9 was discovered in the field and found to exhibit resistance to herbicide concentrations two to three times higher than the recommended dose. Subsequent research identified a Ser_638_Asn mutation in the *BnALS1* gene in M9 [46]. Xin et al. [47] identified two rapeseed materials with seedling-stage tolerance to TBM. Wang et al. [48] screened 241 *Brassica napus* accessions during germination and identified three TBM-tolerant germplasms. Although naturally occurring mutants are valuable for identifying herbicide-tolerant germplasms owing to their regulatory acceptance and research significance, their discovery is largely serendipitous, with low mutation frequencies and limited scalability. As a result, the consistent development of commercially valuable resistant varieties through natural mutation remains challenging.

### 3.2. Development of Herbicide-Resistant Rapeseed Germplasm Through Artificial Mutagenesis

Artificial mutagenesis is an effective approach for the targeted generation of herbicide-resistant mutants, offering a controlled alternative to the inherent unpredictability of natural mutations. This approach has been used extensively in breeding programs worldwide. Commonly used methods include physical and chemical mutagenesis. Physical mutagenesis relies primarily on radiation sources such as X-rays, gamma rays, ultraviolet rays, and lasers. For example, Li et al. [49] irradiated the seeds of the rapeseed cultivar Xiangyou 15 with ^60^Co-γ rays, leading to the identification of eight resistant seedlings through selection on a glufosinate-containing medium. However, compared with chemical mutagenesis, physical mutagenesis induces mutations at a relatively low frequency. For example, the mutation frequency of ^60^Co-γ ray mutagenesis (10^−5^–10^−7^) is significantly lower than that of ethyl methane sulfonate (EMS) mutagenesis (10^−4^–10^−6^). Furthermore, despite its effectiveness, physical mutagenesis often causes significant damage to the organism, reducing both viability and fertility. These drawbacks have limited its practical application and contributed to a decline in its use in more recent studies.

Chemical mutagenesis can be induced by a variety of chemical agents, including sodium azide (AZ), EMS, and methyl-nitrosourea (MNU). Among these, EMS is the most widely used in practical breeding applications. Compared with other mutagens, EMS induces fewer chromosomal aberrations and causes less severe DNA damage, making it a preferred agent for mutagenesis. EMS treatment is relatively simple, requires no specialized equipment, and is cost-effective, enabling simultaneous mutagenesis of large populations and thus improving the efficiency of mutant library construction. Notably, EMS predominantly induces point mutations, often resulting in single-gene alterations with stable phenotypes, which facilitates the screening and identification of desired mutants. In typical protocols, tissues and organs such as pollen, seeds, or microspores are treated with the mutagen, followed by selection on media containing specific herbicides or under field conditions [50,51].

In 1989, Swanson et al. [52] first developed two IMI-resistant rapeseed mutants, PM1 and PM2, through microspore chemical mutagenesis. The PM1 mutant, carrying a Ser_653_Asn mutation in *BnALS1*, exhibited specific resistance to IMI at a recommended field dose of approximately 50 g ha^−1^. By contrast, PM2, which harbored a Trp_574_Leu mutation in *BnALS3*, displayed stronger resistance to IMI (300 g ha^−1^) as well as cross-resistance to SU herbicides. The combination of mutant alleles from PM1 and PM2 further enhanced resistance levels, forming the basis for the development of the commercial ALS-inhibitor-resistant rapeseed cultivar Clearfield. In another study, Tonnemaker et al. [53] generated rapeseed germplasms resistant to a mixture of chlorsulfuron and metsulfuron herbicides through EMS mutagenesis. Hu et al. [54] developed the SU-herbicide-resistant germplasm M342 by introducing a Trp_574_Leu mutation into the *ALS3* gene via EMS mutagenesis. Similarly, Li et al. [55] screened and obtained the M45 mutant, which harbored a Pro_197_Ser substitution in *ALS3*. Lv et al. [56] reported three additional resistant mutants, K1, K4, and K5. K1 and K4 shared the same Pro_197_Ser mutation as M45, whereas K5 carried this mutation in *ALS1* [57]. In addition, the Jiangsu Academy of Agricultural Sciences generated the EMS-induced mutants PN19 and M196. Although these herbicide-resistant germplasms have been successfully identified, many exhibit relatively low levels of resistance, limiting their practical application in breeding programs.

A pyramiding breeding approach was used to develop the highly herbicide-resistant germplasm 5N by combining two EMS-induced resistant lines, PN19 and M342 [20]. The 5N germplasm exhibited a 16-fold increase in resistance relative to the recommended application concentration of the herbicide, conferred by double amino acid substitutions of Trp_574_Leu in both *ALS1* and *ALS3* genes. A second EMS mutagenesis followed by directional selection was then performed on progeny of M342, resulting in the identification of the double-mutant germplasm DS3 [21]. This germplasm carried the *BnALS1* Pro_197_Leu and *BnALS3* Trp_574_Leu mutations and displayed no obvious phytotoxicity when treated with TBM at 12- to 16-fold the recommended concentration. M342, 5N, and DS3 also exhibited resistance to IMI herbicides [54]. Building upon the development of DS3 and 5N, the Jiangsu Academy of Agricultural Sciences successfully bred several non-transgenic SU-resistant rapeseed varieties, including Ning R101, Ning R201, Huinongyou, Jindiyou NO. 1, Su R001 and Hongyou 789, which exhibited resistance to herbicide levels 4-fold higher than the recommended concentration. Likewise, the Hybrid rapeseed research center of Shaanxi Province developed SU-resistant rapeseeds such as Qinyou919R, QinyouR1895, and QinyouR1892 and new SU-resistant varieties developed by the Zhejiang Academy of Agricultural Sciences, Anhui Agricultural University, and Qinghai University are expected to be launched soon. These studies collectively demonstrate that different mutation sites or different amino acid substitutions at the same site can result in significant variation in both the spectrum and level of herbicide resistance. Notably, double mutations often exhibit a resistance-stacking effect. Both natural and artificial mutagenesis have demonstrated that substitutions at Pro_197_, Trp_574_, and Ser_653_ (relative to *Arabidopsis thaliana*) in the *ALS* gene of rapeseed are associated with resistance to ALS inhibitors (Table 1).

### 3.3. Development of Herbicide-Resistant Rapeseed Germplasm by Genetic Engineering

Although EMS mutagenesis can produce a wide range of herbicide-resistant germplasms, it poses several limitations. EMS is both toxic and carcinogenic, raising safety concerns during its application. Moreover, EMS-induced mutations occur randomly, making it challenging to obtain precise multi-base substitutions. Because mutations at the *ALS* gene locus have a cumulative effect on herbicide resistance, single-base changes often produce insufficient resistance, limiting their utility in practical breeding programs. By contrast, genetic engineering offers a safe and controllable method for the development of herbicide-resistant rapeseeds with significant breeding potential. Through the introduction of exogenous resistance genes or the targeted editing of endogenous genes, this strategy enables the development of germplasm with enhanced and stable resistance traits.

#### 3.3.1. Introduction of an Exogenous Resistance Gene

At present, the introduction of exogenous resistance genes in rapeseed is predominantly associated with glyphosate resistance. Class II *EPSPS* enzymes from some microorganisms inherently exhibit high levels of glyphosate resistance. In 1996, Canadian researchers used Agrobacterium-mediated transformation to introduce the *CP4-EPSPS* gene from *Agrobacterium* sp. CP4, together with the modified glyphosate oxidoreductase gene *goxv247*, was introduced into the cultivar Westar to produce the first commercial glyphosate-resistant rapeseed, GT73. Subsequent transgenic lines, such as GT200 (developed using the same strategy) and MON88302 (containing only *CP4-EPSPS*), were also generated. Chinese researchers have similarly generated transgenic rapeseed lines with high glyphosate tolerance by deploying the *CP4-EPSPS*/*gox* gene combination [58]. Upon introduction of the modified *CP4-EPSPS* gene into rapeseed, its protein expression was detectable across various tissues, with the highest levels in leaves, ranging from 6.25 to 21.50 μg/g. This high expression enables the transgenic rapeseed to tolerate glyphosate. Class I EPSPS enzymes are typically sensitive to low concentrations of glyphosate. He et al. [59] obtained the mutated Class I *EPSPS* gene *aroAM12* through genetic engineering. Wang et al. [60] co-transformed *aroAM12* with the *Bacillus thuringiensis* toxin gene *Bts1m*, thereby developing rapeseed lines with both glyphosate and insect resistance. Kahrizi et al. [61] obtained transgenic rapeseed lines tolerant to 10 mM glyphosate (the wild type can tolerate only 1 mM) by performing site-directed mutagenesis on the *Escherichia coli EPSPS* gene (Gly_96_Ala and Ala_183_Thr) and expressing the mutated gene in rapeseed. In addition to EPSPS-based approaches, BASF introduced a mutated *AtALS* gene (Ser_653_Asn) from *Arabidopsis thaliana* to develop the IMI-herbicide-resistant transgenic rapeseed LB-FLFK. Wu et al. [62] isolated the *DsALS-108* gene from a naturally occurring TBM-resistant mutant of *Descurainia sophia* and transferred it into *Brassica napus*, increasing its herbicide resistance approximately threefold.

Transgenic technologies have greatly facilitated the development of herbicide-resistant rapeseed germplasms. However, most currently available resistant rapeseed varieties rely on the introduction of exogenous genes. The commercialization of such transgenic lines has been hampered by ongoing public debates regarding food safety and potential ecological and environmental risks. In light of these challenges, emerging gene-editing technologies offer promising alternatives by enabling the precise modification of endogenous genes to generate non-transgenic herbicide-resistant rapeseed varieties with high breeding value.

#### 3.3.2. Endogenous Gene Editing

The CRISPR/Cas9 gene editing system provides a powerful and precise approach for the creation of herbicide-resistant germplasm [63]. Through targeted nucleotide substitutions or deletions, CRISPR/Cas9 enables the generation of desired phenotypes while avoiding the randomness associated with traditional mutagenesis methods. In a recent study, Wu et al. [64] edited the Pro_197_ site of the *BnALS1* gene in *Brassica napus*, obtaining homozygous Pro_197_Ser mutants (Table 2). These mutants exhibited robust resistance to TBM, tolerating up to three times the recommended field dose without displaying any phytotoxic symptoms. Moreover, the resistance phenotype was stably inherited across generations. Building on this success, the researchers generated *BnALS1*/*BnALS3* double mutants, which showed broader resistance to SU herbicides (Table 2). The precision of CRISPR/Cas9-mediated genome editing not only eliminates the unpredictability of conventional mutagenesis but also avoids the incorporation of exogenous DNA. Notably, the transgenic components used in CRISPR/Cas9 editing can be segregated out through self-pollination or hybridization, resulting in non-transgenic progeny that are phenotypically and genetically indistinguishable from naturally occurring or artificially induced mutants. As a result, several countries, including the United States, Brazil, Argentina, and Canada, have classified CRISPR/Cas9-edited crops as non-transgenic, facilitating their regulatory approval and commercialization [17].

## 4. Future Prospects for Herbicide-Resistant Rapeseed Germplasm Development

The previous section outlined the progression of technologies for the development of herbicide-resistant rapeseed, starting from the identification of natural mutants, through artificial mutagenesis screening and the incorporation of exogenous resistance genes, to the current use of gene-editing technologies. Transgenic herbicide-resistant rapeseed offers simplicity and efficiency, significantly reducing labor costs in production. However, the introduction of foreign genes raises ongoing concerns about biosafety and food safety. Thus, the development of non-transgenic herbicide-resistant rapeseed germplasm is crucial. Initially, this relied on natural mutations and artificial mutagenesis, which are hindered by unpredictability, lengthy genetic improvement cycles, and high labor demands. The availability of CRISPR/Cas9 gene editing now addresses these limitations effectively. As modern technologies continue to evolve, they will be increasingly integrated into plant breeding. Advancing herbicide-resistant rapeseed germplasm must align with these technological developments to explore future prospects in this field (Table 3).

### 4.1. Discovery of New Resistance Genes and Development of New Resistant Rapeseed

Previous studies on herbicide-resistant rapeseed have primarily focused on target-site resistance (TSR) mechanisms, which involve mutations in genes that encode the herbicide’s target enzyme [65]. However, plants can also exhibit non-target-site-resistance (NTSR) mechanisms [66,67,68,69], which reduce the effective concentration of herbicide that reaches the target site. NTSR mechanisms include enhanced herbicide metabolism, sequestration or compartmentalization of herbicides, and reduced absorption and translocation efficiency [70,71,72]. Significant progress in herbicide resistance has also been achieved using NTSR strategies (Table 4). For instance, DuPont created the glyphosate-resistant rapeseed lines 73,496 and 61,061 by introducing the glyphosate N-acetyltransferase gene *gat4621* from *Bacillus licheniformis*. These lines have been widely adopted internationally. Su [73] achieved high glyphosate tolerance in transgenic lines using the glyphosate oxidoreductase gene *gox* from *Achromobacter*. Earlier efforts in developing resistance to glufosinate also laid an important foundation. In 1989, DeBlock et al. [74] pioneered glufosinate-resistant germplasm by introducing the *bar* gene from *Streptomyces hygroscopicus* into rapeseed. In 2015, Zhang [75] integrated the *bar* gene into the cultivars Xiangyou 15 and 742R, thus enhancing glufosinate resistance. Bayer also developed the widely used glufosinate-resistant rapeseed cultivar T45 by introducing the *pat* gene from *Streptomyces viridochromogenes*.

In addition to exogenous resistance genes derived from microorganisms, detoxification genes from plants can also be used in the development of resistant germplasm. Genes involved in metabolic detoxification pathways, such as cytochrome P450 monooxygenases (CYP450s), glycosyltransferases (GTs), and glutathione-S-transferases (GSTs), have been implicated in herbicide resistance of both weeds and cultivated crops [76,77,78,79,80,81,82,83]. Future research should focus on investigating the roles of these gene families in *Brassica napus* to develop novel herbicide-resistant rapeseed varieties. Ideally, endogenous herbicide-resistance genes from these families can be identified and overexpressed in rapeseed to enhance metabolic resistance. The discovery and functional characterization of such endogenous resistance genes will rely heavily on the integration of multi-omics technologies and the application of artificial intelligence (AI) to process and analyze large-scale omics datasets.

### 4.2. Integration of Multi-Omics and Artificial Intelligence for Precision Breeding

The integration of multi-omics technologies, including genomics, transcriptomics, and metabolomics, is a powerful strategy for systematic characterization of the regulatory networks that underlie resistance mechanisms and key agronomic traits. For example, Zhang et al. [84] used Hi-C and ATAC-seq to analyze high-oil and low-oil rapeseed lines. By combining these datasets with transcriptomic, genomic, and QTL fine-mapping analyses, they revealed the three-dimensional (3D) spatial structure of the genome and gained insight into the molecular mechanisms by which *BnaA09g48250D* regulates seed oil content. Zhang et al. [85] constructed a pan-genome for *Brassica napus* by integrating 16 reference genomes with resequencing data from 2105 core germplasm accessions. They developed a joint analytical framework that combined genome-wide association studies (GWAS), eQTL, transcriptome-wide association studies (TWAS), and colocalization analyses to construct regulatory networks for complex agronomic traits. Furthermore, Hu et al. [86] conducted high-depth resequencing of 418 modern rapeseed accessions to investigate genomic selection patterns and the genetic structure of agronomic traits. Their results indicated that rapeseed breeding in China has progressed through two major phases: initial selection for environmental adaptation, followed by optimization for high yield and quality.

AI is driving transformative innovations in plant breeding paradigms (Figure 4) [87]. By leveraging deep-learning models such as AlphaFold3, the 3D structures of target proteins and their herbicide-binding affinities can be accurately predicted, facilitating virtual screening of candidate mutations for optimal resistance and accelerating the discovery of effective resistance alleles [88,89]. Convolutional neural networks (CNN) can automate the analysis of hyperspectral field images, enabling high-throughput, non-destructive assessment and correlation of resistance phenotypes (e.g., herbicide damage and recovery) with key agronomic traits (e.g., biomass and silique number), thereby expediting the development of superior germplasm [90,91]. Zhang et al. [92] demonstrated the potential of such technologies by using a high-throughput phenotyping platform that integrated hyperspectral and RGB imaging to monitor phenotypic responses in a natural population of 505 *Brassica napus* accessions under normal, low-salt-stress, and high-salt-stress conditions. GWAS analysis identified numerous candidate genes and QTLs associated with salt stress response, including *BnERF3*, whose function in salt tolerance was validated through transgenic experiments, demonstrating the utility of combining imaging, genomics, and functional validation. Machine learning (ML) and deep learning (DL) approaches are also increasingly used to improve parent selection, predict heterosis, and refine genomic selection models, enhancing decision-making efficiency in herbicide resistance breeding programs [93,94]. The integration of AI with multi-omics, high-throughput phenotyping, and genome editing is poised to revolutionize the breeding of herbicide-resistant rapeseed.

### 4.3. Exploring New Strategies for Synthetic Biology and Directed Evolution

Synthetic biology offers powerful tools for crop improvement by enabling the systematic design and modification of biological systems on the basis of engineering principles. This iterative process follows a “design-build-test-learn” cycle and holds great promise for enhancing agronomic traits and biological yield in crops (Figure 5) [95]. Through the integration of regulatory elements and biosensors designed with AI, modified crops can detect a wide range of plant infections (including fungi, bacteria, and viruses), plant hormone fluctuations, and responses to abiotic stresses. These signals can then be translated into quantifiable outputs, enabling the reprogramming of cell, tissue, and organ development to rapidly achieve desired plant morphologies and functions [96]. By constructing herbicide-activated, promoter-regulated expression systems for resistance genes, it is possible to precisely control resistance so that it is induced only upon herbicide application. This approach mitigates the metabolic burden associated with constitutive expression of resistance genes and minimizes adverse effects on crop growth. In addition, engineering rhizosphere probiotics to secrete specific herbicide-degrading enzymes can enhance the capacity of the rhizosphere to degrade herbicides. This “plant–microbe” collaborative strategy not only reduces soil herbicide residues but also reduces the requirement for high expression of endogenous resistance genes in plants, offering new possibilities for developing environmentally friendly herbicide-resistant rapeseed.

Directed evolution is a set of techniques that mimic natural selection by artificially introducing gene mutations and screening for desired post-mutation characteristics, thereby generating biomolecules (e.g., enzymes, proteins, and metabolic pathways) with specific functions or improved properties. Among these approaches, directed protein evolution has become a primary focus of current research. For example, Xu et al. [97] performed evolutionary and machine learning analyses of the amino acid sequences of Coq1, a key enzyme in the coenzyme Q synthesis pathway, across more than 1000 terrestrial plants. Using CRISPR/Cas9-mediated gene editing, they introduced five targeted amino acid substitutions and successfully enabled the biosynthesis of coenzyme Q_10_ in rice. Noam Prywes et al. [98] constructed a nearly complete Rubisco mutant library, encompassing 8760 single-amino-acid substitutions, and used engineered *E. coli* for high-throughput screening. This approach led to the identification of multiple mutation sites that enhance the affinity of Rubisco for carbon dioxide and revealed the complex relationship between the function of Rubisco and its amino acid sequence. In addition, Wang et al. [99] developed the *Arabidopsis* CBE editing vector pHEE901-A3A using the efficient A3A-PBE editor under the oocyte-specific EC1 promoter. Using this system, they performed high-throughput directed evolution of key herbicide target genes, ultimately obtaining the glyphosate-tolerant EPSPS mutant T178I/A179V/P182S and 46 ALS-herbicide-resistant mutants. They also established the largest single-gene plant mutant library for the 4-hydroxyphenylpyruvate dioxygenase (HPPD) gene to date.

## 5. Potential Challenges and Solutions for the Promotion of Herbicide-Resistant Rapeseed

The widespread cultivation of herbicide-resistant rapeseed poses several ecological risks, primarily associated with gene flow. As a cross-pollinated species, rapeseed is capable of naturally hybridizing with wild relatives, thereby facilitating the transfer of resistance genes into wild or weedy populations [100,101,102]. In addition, the prolonged and repeated use of herbicides with a single mode of action can accelerate the evolution of herbicide resistance in weed species and lead to the emergence of volunteer rapeseed plants [103,104,105,106,107,108,109,110,111,112]. According to recent data from the International Survey of Herbicide-Resistant Weeds, more than 70 weed species have developed resistance to glyphosate, and 176 species are now resistant to ALS inhibitors worldwide [113].

To mitigate the risk of gene flow, a multifaceted prevention and control strategy should be adopted. From a biological perspective, developing and deploying male-sterile lines can effectively prevent pollen dispersal, and leveraging maternal inheritance traits can further restrict gene escape. In terms of agricultural management practices, establishing physical-isolation buffer zones around planting areas and implementing crop rotation with non-cruciferous species can significantly reduce the probability of hybridization. In addition, pyramiding resistance genes with distinct modes of action to develop multi-resistant rapeseed varieties, coupled with the alternating use of herbicides with different modes of action, can delay the evolution of weed resistance and reduce the emergence of volunteer plants. Furthermore, integrating Internet of Things (IoT) sensors and AI-based image-recognition technologies can facilitate the development of intelligent pesticide application systems. These systems can monitor the weed community (e.g., species, density) in real-time, dynamically optimize the pesticide combination and application scheme, and achieve precise variable-rate pesticide application. Consequently, this approach minimizes the selection pressure exerted by herbicides, effectively delaying the emergence and spread of herbicide-resistant weeds.

## 6. Conclusions

The development of herbicide-resistant rapeseed represents a significant advance in addressing weed-related damage, reducing production costs, and promoting high-quality rapeseed production. The development of herbicide-resistant rapeseed germplasm holds significant practical significance. It effectively addresses the problem of weed damage: non-transgenic varieties such as “Ning R101” have significantly improved weed control efficiency and crop yield, saving 60–70 yuan in labor costs per mu and increasing growers’ income. Meanwhile, it makes outstanding contributions to sustainable agriculture. The supporting herbicides reduce pesticide usage by 30% and minimize soil disturbance caused by mechanical weeding. Over the past few decades, various approaches, including natural variation, artificial mutagenesis and transgenic technology, have enabled the development of numerous excellent resistant germplasms and the cultivation of resistant varieties for widespread use. The increasing maturity and integration of gene-editing and multi-omics technologies have significantly enhanced both the precision and efficiency of herbicide-resistant rapeseed development, promoting the mechanization process of the rapeseed industry. Looking ahead, emerging fields and technologies such as synthetic biology, directed protein evolution and AI are expected to further optimize the design and development of resistant germplasm. These technological advances will ultimately promote the efficient and sustainable growth of global rapeseed production. Nevertheless, it is essential to maintain vigilant risk management throughout this process, particularly with regard to the potential gene flow of resistance traits, the evolution of weed resistance, and the control of herbicide-resistant volunteer plants.

## Figures and Tables

**Figure 1 ijms-26-08292-f001:**
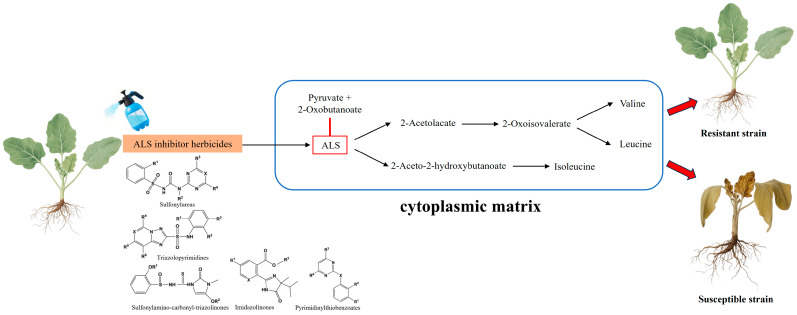
Molecular mechanisms of the ALS-inhibiting herbicides.

**Figure 2 ijms-26-08292-f002:**
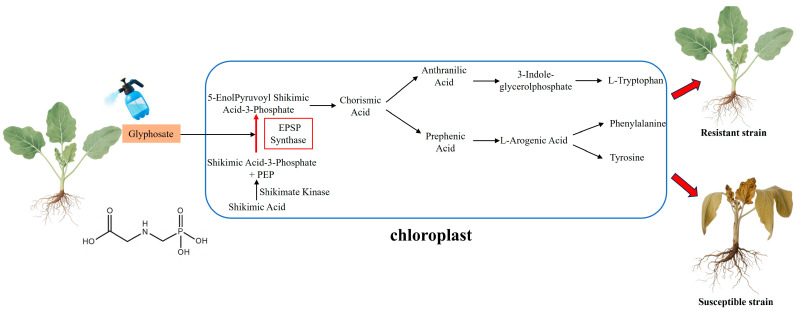
Molecular mechanisms of glyphosate.

**Figure 3 ijms-26-08292-f003:**
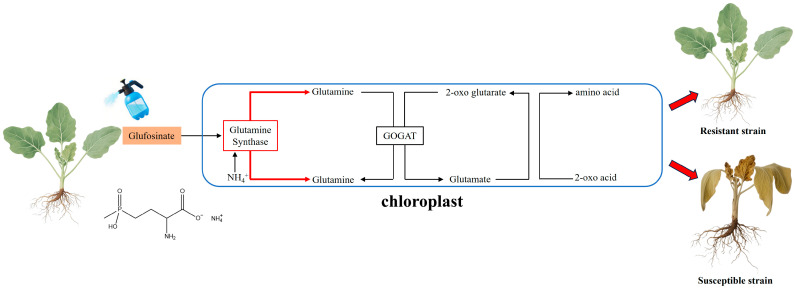
Molecular mechanisms of glufosinate.

**Figure 4 ijms-26-08292-f004:**
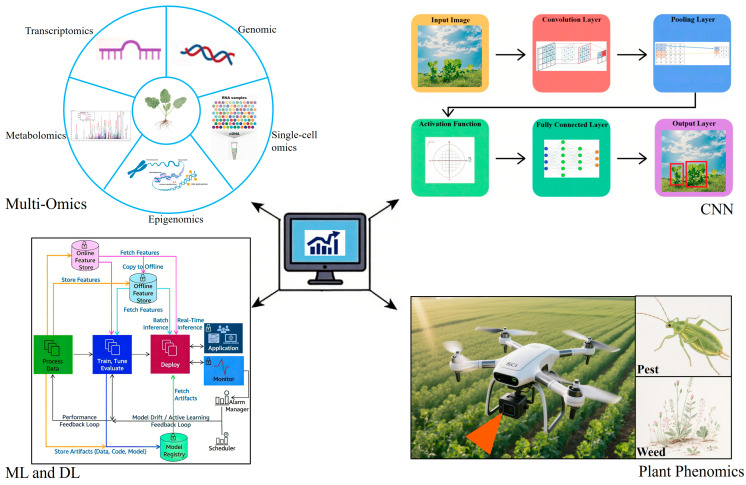
Multi-omics and AI-driven plant breeding.

**Figure 5 ijms-26-08292-f005:**
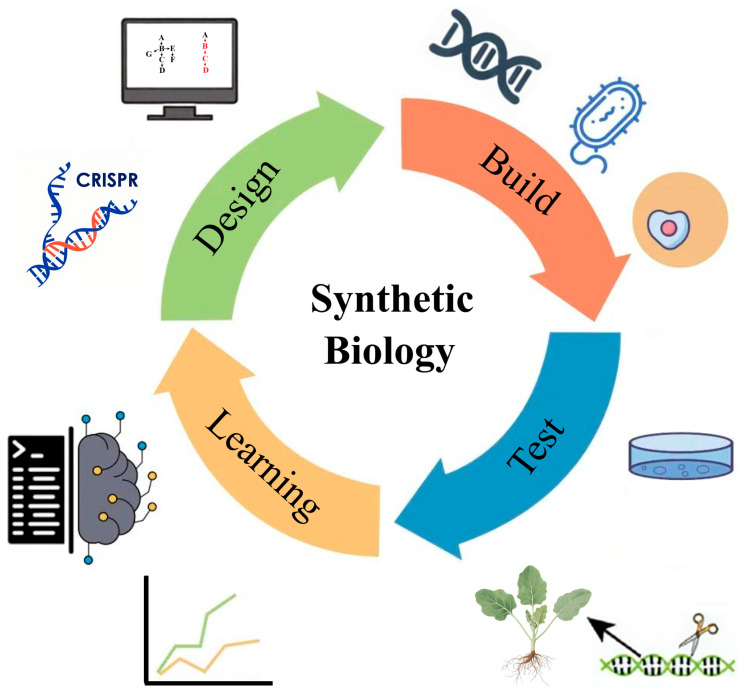
Intelligent breeding approach in synthetic biology.

**Table 1 ijms-26-08292-t001:** Rapeseed germplasm resources with resistance to ALS inhibitor herbicides obtained through natural or artificial mutation.

Materials	Methods	Genes	Herbicides Resistance	Amino Acid Resistant Mutations	Resistance Level Data	Reference
PM1	ENU	*ALS1*	IMI	Ser_653_Asp	50 g ha^−1^	[52]
PM2	ENU	*ALS3*	IMI, SU, TP	Trp_574_Leu	300 g ha^−1^	[52]
M9	Spontaneous mutation	*ALS1R*	IMI	Ser_653_Asn	90 g ha^−1^	[46]
PN19	EMS	*ALS1*	SU, SCT	Trp_574_Leu	50 g ha^−1^	[20]
M342	EMS	*ALS3R*	SU	Trp_574_Leu	90 g ha^−1^	[54]
5N	M342 × PN19	*ALS1*	SU	Trp_574_Leu	360 g ha^−1^	[20]
		*ALS3*				
K5	EMS	*ALS1*	SU	Pro_197_Ser	15 g ha^−1^	[57]
DS3	EMS	*ALS1*	SU	Pro_197_Ser	270–360 g ha^−1^	[21]
		*ALS3*		Trp_574_Leu		
12WH318	Introduction	*ALS1*	IMI	Ser_653_Asp	72–108 g ha^−1^	
M45, K1, K4	EMS	*ALS3*	SU	Pro_197_Ser/Leu	3, 0.6, 0.6 g ha^−1^	[56]
ALS1 mutation	In vitro mutation	*ALS1*	IMI	Ser_653_ deletion	-	
RCS-5	Spontaneous mutation	-	IMI, SU	-	10 g ha^−1^	
M-37, M-42	ENU	-	SU	-	30 g ha^−1^	

Note: The sites of resistance mutations (amino acids or bases) are based on the *Arabidopsis thaliana* genome.

**Table 2 ijms-26-08292-t002:** Herbicide-resistant rapeseed germplasm resources obtained through genetic modification.

Materials	Genes	Transgenic Source	Herbicides Resistance	Reference
CT73/GT200	*CP4-EPSPS*+*goxv247*	*Agrobacterium* sp. *CP4+Pseudomonas* sp.	Glyphosate	[58]
MON88302	*CP4-EPSPS*	*Agrobacterium* sp. *CP4*	Glyphosate	[58]
XT	*aroAM12*	*Salmonella typhimurium*	Glyphosate	[59]
LB-FLFK	*AtALS*	*Arabidopsis thaliana*	IMI	[61]
DsALS-108-26	*DsALS-108*	*Descurainia sophia*	TBM	[62]
P197S mutants	*BnALS1*	*Brassica napus*	TBM	[64]
	*BnALS1/BnALS3*	*Brassica napus*	TBM	[64]

**Table 3 ijms-26-08292-t003:** Development of herbicide-resistant germplasm breeding strategies for rapeseed.

Development Stage	Breeding Strategies	Benefits	Challenges
First	Natural mutation	Germplasm can be directly used	Un-predictability
Second	Artificial mutation	Targeted improvement of a single trait	Lengthy genetic improvement cycles
Third	Genetic modification	Simplicity and efficiency	Introduction of foreign genes
	Gene editing	High efficiency and short cycle	Off-target risk
Fourth	Artificial intelligence	Efficient and intelligent	The technology is still immature

**Table 4 ijms-26-08292-t004:** Development of herbicide-resistant rapeseed germplasm using NTSR mechanisms.

Materials	Genes	Transgenic Source	Herbicides Resistance	Reference
61061/73496	*gat4621*	*Bacillus licheniformis*	Glyphosate	
Transgenic plant	*gox*	*Pseudomonas* sp.	Glyphosate	[73]
MS8	*bar*	*Streptomyces hygroscopicus*	Glufosinate	
RF1	*bar*	*Streptomyces hygroscopicus*	Glufosinate	
MS1	*bar*	*Streptomyces hygroscopicus*	Glufosinate	
PGS1	*bar*	*Streptomyces hygroscopicus*	Glufosinate	
HCN10	*bar*	*Streptomyces hygroscopicus*	Glufosinate	
PHY14	*bar*	*Streptomyces hygroscopicus*	Glufosinate	
15A	*bar*	*Streptomyces hygroscopicus*	Glufosinate	
742R	*bar*	*Streptomyces hygroscopicus*	Glufosinate	[63]
T45	*pat*	*Streptomyces viridochromo*	Glufosinate	[63]

## Data Availability

Not applicable.

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
