# Peer review of "Molecular Mechanisms of Herbicide Resistance in Rapeseed: Current Status and Future Prospects for Resistant Germplasm Development"

_ijms, 2025, doi:10.3390/ijms26178292_

Round 1
Reviewer 1 Report
Comments and Suggestions for Authors
To authors,
This manuscript presents a review of the mode of action of herbicides on plants. The article presents relevant figures, with metabolic pathways related to the mode of herbicide action and consequences for the target plant. The article presents future perspectives and examples using model plants. The article's writing and presentation comply with the journal's standards.
Author Response
This manuscript presents a review of the mode of action of herbicides on plants. The article presents relevant figures, with metabolic pathways related to the mode of herbicide action and consequences for the target plant. The article presents future perspectives and examples using model plants. The article's writing and presentation comply with the journal's standards.
Response: Thank you very much for your positive evaluation and constructive comments on our manuscript. We greatly appreciate your careful reading and valuable feedback, which have provided important encouragement for our work.
Reviewer 2 Report
Comments and Suggestions for Authors
The manuscript by Liu et al. titled “Molecular Mechanisms of Herbicide Resistance in Rapeseed: Current Status and Future Prospects for Resistant Germplasm Development” is a well-structured and thoroughly researched review that makes a valuable contribution to the field of crop molecular biology. The writing is clear, technically sound, and logically organized, with each section building coherently on the last—from the biochemical modes of action of major herbicides to the molecular basis of resistance and the breeding approaches applied in rapeseed. The figures and tables are of excellent quality and well-integrated into the text. In particular, schematic illustrations clearly depict resistance mechanisms, and the tables provide concise summaries of resistant germplasm lines, target genes, and mutation profiles, enhancing the manuscript’s utility for researchers and breeders alike. The review successfully captures the current landscape of herbicide resistance in rapeseed and outlines future directions, including the integration of multi-omics, AI, and synthetic biology. While the focus on glyphosate, glufosinate, and ALS inhibitors is well justified, the manuscript would benefit from briefly acknowledging earlier herbicide resistance systems—such as triazine-tolerant rapeseed—to provide historical context. Additionally, a short comparative analysis of the different breeding strategies (natural mutation, chemical mutagenesis, transgenic approaches, and CRISPR/Cas9 editing) would help synthesize the presented information and guide the reader in understanding their relative strengths and limitations. With these enhancements, the manuscript will serve as an even more complete and authoritative reference on the development of herbicide-resistant rapeseed and is highly recommended for publication
Author Response
The manuscript by Liu et al. titled “Molecular Mechanisms of Herbicide Resistance in Rapeseed: Current Status and Future Prospects for Resistant Germplasm Development” is a well-structured and thoroughly researched review that makes a valuable contribution to the field of crop molecular biology. The writing is clear, technically sound, and logically organized, with each section building coherently on the last-from the biochemical modes of action of major herbicides to the molecular basis of resistance and the breeding approaches applied in rapeseed. The figures and tables are of excellent quality and well-integrated into the text. In particular, schematic illustrations clearly depict resistance mechanisms, and the tables provide concise summaries of resistant germplasm lines, target genes, and mutation profiles, enhancing the manuscript’s utility for researchers and breeders alike. The review successfully captures the current landscape of herbicide resistance in rapeseed and outlines future directions, including the integration of multi-omics, AI, and synthetic biology. While the focus on glyphosate, glufosinate, and ALS inhibitors is well justified, the manuscript would benefit from briefly acknowledging earlier herbicide resistance systems-such as triazine-tolerant rapeseed-to provide historical context. Additionally, a short comparative analysis of the different breeding strategies (natural mutation, chemical mutagenesis, transgenic approaches, and CRISPR/Cas9 editing) would help synthesize the presented information and guide the reader in understanding their relative strengths and limitations. With these enhancements, the manuscript will serve as an even more complete and authoritative reference on the development of herbicide-resistant rapeseed and is highly recommended for publication.
Response: Thank you sincerely for your insightful comments and constructive suggestions on our manuscript. 1. We fully agree with your observation that incorporating earlier herbicide resistance systems—such as triazine-tolerant rapeseed—would enrich the historical context of our review. In the revised manuscript, we have added relevant descriptions of such resistance systems. We have made modifications to the Introduction, specifically at lines 62–64 on page 2. In addition, we fully agree with the value of comparing different breeding strategies, and we have added this information to the revised version. These modifications are found specifically at lines 353–364 on page 11.
Once again, we deeply appreciate your time and expertise in evaluating our work. Your guidance has been instrumental in strengthening the scientific rigor and comprehensiveness of our review. We look forward to submitting the revised manuscript and hope that it will now meet the journal’s standards.
Reviewer 3 Report
Comments and Suggestions for Authors
Review Report
Title of the manuscript: “Molecular Mechanisms of Herbicide Resistance in Rapeseed: Current Status and Future Prospects for Resistant Germplasm Development”
This review aims to summarize current studies on the mechanisms of action and plant resistance associated with herbicide use, as well as the development of herbicide-resistant rapeseed germplasm, and to discuss future research and potential challenges associated with the application of herbicide-resistant rapeseed. Although this review provides an interesting overview for understanding molecular mechanisms of herbicide resistance in rapeseed, several aspects need to be improved to improve the quality of the manuscript:
- Title:
The addition of the herbicides described in the mechanism of action (glyphosate, glufosinate, and acetolactate synthase) may add sufficient precision to the title.
- Abstract
- Please add some examples of the mode of action reported in the review in the abstract section.
- Please put the keywords in alphabetical order.
- Introduction
- Please remove the FAO link in the introduction and add it as a digital citation and as a full reference in the reference list.
- Please explain precisely what this "considerable economic importance" of rapeseed cultivation is. And start with a new paragraph when you discuss weeds.
- Why do you specify "traditional weed control in Chinese rapeseed fields" rather than in other countries?
- Several sentences and important information do not have any bibliographic references. Please add recent and relevant bibliographic references.
- Please mention the strength or the gap filled by this review compared to other existing literature in this direction.
- Modes of action of three major herbicides and the molecular mechanisms of crop resistance
- Before moving on to section “2.1 ALS and ALS Inhibiting Herbicides”, please add a short paragraph on the importance of this section (2. Modes of Action, ...) and their objectives.
- Please improve the visual quality of all figures.
- Please remove the titles of the figures, especially "Working mechanisms", and replace them with a more appropriate scientific word.
- Please discuss where exactly these mechanisms take place at the cellular level. This can also be added to the figures.
- Research progress in the development of herbicide-resistant rapeseed Germplasm
- Before moving on to section “3.1 Development of herbicide-resistant…”, please add a short paragraph on the importance of this section (5. Research progress ...) and its objectives.
- Please add a column for references in Tables 1 and 2.
- It will be interesting to add a summary table on the studies carried out in the direction of progress in the development of herbicide-resistant rapeseed germplasm.
- Future prospects for herbicide-resistant rapeseed germplasm development
- Before moving on to section “4.1 Discovery of new genes…”, please add a short paragraph on the importance of this section (4. Future prospects ...) and its objectives.
- Please add a column for references in Table 3.
- Please add a table summarizing the benefits, challenges, and potential solutions in the development of herbicide-resistant rapeseed germplasm and for other innovative solutions (integration of multi-omics and artificial intelligence for precision breeding, etc.).
- Please highlight the practical implications of herbicide-resistant rapeseed germplasm development in the real world, particularly in terms of their potential contribution to sustainable agriculture.
- Citation and References
Several paragraphs need to be referenced. Please write the references correctly using a unique format based on the MDPI style.
Best regards.
Author Response
- Title:
The addition of the herbicides described in the mechanism of action (glyphosate, glufosinate, and acetolactate synthase) may add sufficient precision to the title.
Response 1: Thank you very much for your suggestions. Although our review mainly mentions three types of herbicides (glyphosate, glufosinate, and acetolactate synthase-inhibiting herbicides), there are also a small number of rapeseed germplasms exhibiting resistance to other types of herbicides with poor understanding of molecular mechanisms (lines 61–64 on page 2) . Therefore, specific herbicide types are not indicated in the title of this paper.
- Abstract
- Please add some examples of the mode of action reported in the review in the abstract section.
- Please put the keywords in alphabetical order.
Response 2: Your suggestions are appreciated. However, due to the word limit, we omitted detailed examples of the mechanism of action. The keywords have been adjusted as requested (lines 32 on page 1).
- Introduction
- Please remove the FAO link in the introduction and add it as a digital citation and as a full reference in the reference list.
- Please explain precisely what this "considerable economic importance" of rapeseed cultivation is. And start with a new paragraph when you discuss weeds.
- Why do you specify "traditional weed control in Chinese rapeseed fields" rather than in other countries?
- Several sentences and important information do not have any bibliographic references. Please add recent and relevant bibliographic references.
- Please mention the strength or the gap filled by this review compared to other existing literature in this direction.
Response 3: We have deleted and revised the FAO link as well as the content in the introduction that you raised objections to, with the specific revisions located in lines 36-39 on page 1. For some sentences, the cited references are derived from Chinese-language literature, so we have not included them in the main text. The same applies to the Table below where no references were added. Compared with other reviews in the same direction, the unique innovation of this review lies in that after summarizing the methods for creating herbicide-resistant germplasms, the authors further discuss the future approaches to herbicide-resistant germplasm creation, providing references for breeders.
- Modes of action of three major herbicides and the molecular mechanisms of crop resistance
- Before moving on to section “2.1 ALS and ALS Inhibiting Herbicides”, please add a short paragraph on the importance of this section (2. Modes of Action, ...) and their objectives.
- Please improve the visual quality of all figures.
- Please remove the titles of the figures, especially "Working mechanisms", and replace them with a more appropriate scientific word.
- Please discuss where exactly these mechanisms take place at the cellular level. This can also be added to the figures.
Response 4: We agree with the revision suggestions you put forward. We have added relevant content before section 2.1 (lines 88-92 on page 4) and revised the title of the figure. In addition, we have indicated the cellular locations where the known mechanisms occur in the figure and improved the quality of the figure (lines 103 on page 4, lines 136 on page 5, and lines 173 on page 6).
- Research progress in the development of herbicide-resistant rapeseed Germplasm
- Before moving on to section “3.1 Development of herbicide-resistant…”, please add a short paragraph on the importance of this section (5. Research progress ...) and its objectives.
- Please add a column for references in Tables 1 and 2.
- It will be interesting to add a summary table on the studies carried out in the direction of progress in the development of herbicide-resistant rapeseed germplasm.
Response 5: We have added relevant content before section 3.1 (lines 188-193 on page 7) and supplemented the references for Table 1 and Table 2. Additionally, we have also integrated the two items into Table 3 based on the 6th suggestion you put forward. The specific content is located in lines 366-367 on page 12.
- Future prospects for herbicide-resistant rapeseed germplasm development
- Before moving on to section “4.1 Discovery of new genes…”, please add a short paragraph on the importance of this section (4. Future prospects ...) and its objectives.
- Please add a column for references in Table 3.
- Please add a table summarizing the benefits, challenges, and potential solutions in the development of herbicide-resistant rapeseed germplasm and for other innovative solutions (integration of multi-omics and artificial intelligence for precision breeding, etc.).
- Please highlight the practical implications of herbicide-resistant rapeseed germplasm development in the real world, particularly in terms of their potential contribution to sustainable agriculture.
Response 6: In accordance with your requirements, we have added relevant content before section 4.1 (lines 353-364 on page 11) and supplemented the references for Table 4. Furthermore, we have emphasized the practical significance of herbicide-resistant rapeseed germplasm development in the real world as well as its potential contributions to sustainable agriculture in the conclusion section. The specific content is located in lines 498-503 on page 16.
- Citation and References
Several paragraphs need to be referenced. Please write the references correctly using a unique format based on the MDPI style.
Response 7: We have further corrected and revised the references to make them conform to the style requirements of MDPI journals.
Reviewer 4 Report
Comments and Suggestions for Authors
Dear authors,
Your article "Molecular Mechanisms of Herbicide Resistance in Rapeseed: Current Status and Future Prospects for Resistant Germplasm Development "systematically reviews the molecular mechanisms, germplasm creation progress, and future research directions of herbicide resistance in rapeseed, covering three major categories of herbicides: ALS inhibitors, glyphosate, and glufosinate ammonium. It involves natural mutagenesis, artificial mutagenesis, genetic engineering, and gene editing technologies, and the content is comprehensive and forward-looking. However, the paper still needs to be revised with the following issues before publication:
1.Abstract: It is necessary to clarify the "unique contributions" of the review, such as "the first systematic integration of non target resistance (NTSR) mechanism and multi omics/AI technology in rapeseed resistance breeding", and streamline the description of existing technologies (such as traditional mutagenesis), highlighting the innovation of cutting-edge directions such as CRISPR and synthetic biology.
2.introduction: Add "current research pain points" such as "insufficient analysis of non target resistance mechanisms, leading to delayed creation of multi resistance germplasm" and "lack of systematic data on off target risk assessment of gene editing technology in rapeseed" to highlight the necessity of the review.
3.Add "resistance level data" (such as the resistance multiple of PM1 to IMI) to Table 1, and label the field application performance of each mutant (such as "Clearfield variety has a weed control effect of over 90% in the Yangtze River Basin of China").
3.Comparing the efficiency of different mutagenesis methods: for example, the mutation frequency of EMS mutagenesis (10 ⁻⁴ -10 ⁻⁶) is significantly higher than that of ⁶⁰ Co - γ rays (10 ⁻⁵ -10 ⁻⁷), but it can easily lead to decreased fertility.
4.ALS inhibitor section: Clarify the specific representative agents of the "5 chemical structures" (SU, IMI, etc.) (such as SU class chlorsulfuron) and their practical application scenarios in rapeseed fields.
5.Glyphosate section: Supplement the expression efficiency data of Class II EPSPS (such as CP4-EPSPS) in rapeseed (such as "the enzyme activity of CP4-EPSPS in transgenic rapeseed is 2-3 times higher than that of endogenous EPSPS") to enhance the empirical explanation of the mechanism.
6.Supplement the latest research from 2024-2025, such as "NTSR mechanism of BnABC transporter mediated glyphosate efflux published in 2024" and "Efficient editing case of CRISPR-Cas12a in rapeseed in 2025".
7.Unified terminology: The full text of "ALS" and "AHAS" should be consistent (it is recommended to unify them as ALS)
Comments on the Quality of English LanguageThe English could be improved to more clearly express the research.
Author Response
Dear authors,
Your article "Molecular Mechanisms of Herbicide Resistance in Rapeseed: Current Status and Future Prospects for Resistant Germplasm Development "systematically reviews the molecular mechanisms, germplasm creation progress, and future research directions of herbicide resistance in rapeseed, covering three major categories of herbicides: ALS inhibitors, glyphosate, and glufosinate ammonium. It involves natural mutagenesis, artificial mutagenesis, genetic engineering, and gene editing technologies, and the content is comprehensive and forward-looking. However, the paper still needs to be revised with the following issues before publication:
- Abstract: It is necessary to clarify the "unique contributions" of the review, such as "the first systematic integration of non target resistance (NTSR) mechanism and multi omics/AI technology in rapeseed resistance breeding", and streamline the description of existing technologies (such as traditional mutagenesis), highlighting the innovation of cutting-edge directions such as CRISPR and synthetic biology.
Response 1: Thank you for pointing this out. We agree with this comment and have made relevant revisions (lines 17–27 on page 1).
- Introduction: Add "current research pain points" such as "insufficient analysis of non target resistance mechanisms, leading to delayed creation of multi resistance germplasm" and "lack of systematic data on off target risk assessment of gene editing technology in rapeseed" to highlight the necessity of the review.
Response 2: We agree with this suggestion and have modified the Introduction accordingly (lines 67–76 of page 2).
- Add "resistance level data" (such as the resistance multiple of PM1 to IMI) to Table 1, and label the field application performance of each mutant (such as "Clearfield variety has a weed control effect of over 90% in the Yangtze River Basin of China").
Response 3: Thank you for this suggestion; we have added resistance-level data to Table 1. However, because the rapeseed germplasm in Table 1 has not yet been planted in large-scale field trials, we do not have data on field application performance.
- Comparing the efficiency of different mutagenesis methods: for example, the mutation frequency of EMS mutagenesis (10⁻⁴ -10⁻⁶) is significantly higher than that of ⁶⁰Co-γ rays (10⁻⁵ -10⁻⁷), but it can easily lead to decreased fertility.
Response 4: Thank you; this is useful information, and we have added it at lines 236–238 on page 8.
- ALS inhibitor section: Clarify the specific representative agents of the "5 chemical structures" (SU, IMI, etc.) (such as SU class chlorsulfuron) and their practical application scenarios in rapeseed fields.
Response 5: Sulfonylureas, such as tribenuron-methyl, effectively manage broad-leaved weeds like Descurainia sophia, Capsella bursa-pastoris, and Galium aparine. Imidazolinones, exemplified by imazethapyr, target weeds such as Amaranthus spp., Abutilon theophrasti, Solanum nigrum, and Xanthium sibiricum. Triazolopyrimidines, of which florasulam is a representative, control weeds such as Galium aparine, Descurainia sophia, and Capsella bursa-pastoris. Pyrimidinyl benzoates like bispyribac-sodium are effective against Monochoria vaginalis, Equisetum ramosissimum, and Sagittaria trifolia. Sulfonylamino-carbonyl-triazolinones, such as triflumizone, manage weeds including Galium aparine, Descurainia sophia, Capsella bursa-pastoris, and Stellaria media. However, because these herbicides can also kill rapeseed, their use in rapeseed fields is limited. Although these herbicides can be used on their corresponding resistant germplasms or varieties (such as those in Table 1), they cannot be used on ordinary varieties, and there are thus no practical application scenarios in rapeseed fields.
- Glyphosate section: Supplement the expression efficiency data of Class II EPSPS (such as CP4-EPSPS) in rapeseed (such as "the enzyme activity of CP4-EPSPS in transgenic rapeseed is 2-3 times higher than that of endogenous EPSPS") to enhance the empirical explanation of the mechanism.
Response 6: We agree, and we have made your suggested revisions at lines 313–316 on page 10.
- Supplement the latest research from 2024-2025, such as "NTSR mechanism of BnABC transporter mediated glyphosate efflux published in 2024" and "Efficient editing case of CRISPR-Cas12a in rapeseed in 2025".
Response 7: Thank you for suggesting this relevant literature on the latest research. We are very sorry that we could not find the two articles you listed and were unfortunately unable to add them to the article.
- Unified terminology: The full text of "ALS" and "AHAS" should be consistent (it is recommended to unify them as ALS)
Response 8: Thank you for noting this issue; we have updated all instances to ALS for consistency.
Reviewer 5 Report
Comments and Suggestions for Authors
I found this review to be well written, easy to read and informative. The standard of writing and the logical flow were good, and although I am not an expert in herbicide resistance per se, the coverage of the main molecular mechanisms of resistance appeared to be adequate. The current use of CRISPR techniques was explained, and the review finishes with a more forward looking, speculative overview of the currently employed research techniques and future technological approaches to understanding and improving resistance, although th projected gains using this techniques are (perhaps necessarily) rather vague. I did have a question that did not appear to be addressed in detail. Given that CRISPR modification can create non-GM resistant strains (via point mutations) with excellent resistance to each herbicide, the authors do not explain the motivation for exploring other indirect mechanisms of resistance, other than an improved understanding of plant physiology. Is there a potential disadvantage of CRISPR engineering that means that this approach may become unsuitable in future? Please address this point.
Other topics that I would have liked to have seen were (1) a projection of the likely future of herbicide resistance technology for weed control given that the use of herbicides (though necessary at present) is fundamentally undesirable in terms of their ecological effects. For example, the use of RNAi approaches to specific weed control (similar questions can be asked about pest resistance of course. (2) What other upcoming candidate herbicides are being explored (other than the three classes discussed) and what might their prospects be? (3) What impacts do current approaches such as transgenics have on plant yield and viability? (4) How do herbicide use, pest control, fertiliser use and the soil microbiome integrate in real crop scenarios and what is known of the effect of herbicides on these other aspects of cropping? A detailed discussion is not needed but some pointers to the wider literature would be appreciated.
Author Response
I found this review to be well written, easy to read and informative. The standard of writing and the logical flow were good, and although I am not an expert in herbicide resistance per se, the coverage of the main molecular mechanisms of resistance appeared to be adequate. The current use of CRISPR techniques was explained, and the review finishes with a more forward looking, speculative overview of the currently employed research techniques and future technological approaches to understanding and improving resistance, although the projected gains using this techniques are (perhaps necessarily) rather vague. I did have a question that did not appear to be addressed in detail. Given that CRISPR modification can create non-GM resistant strains (via point mutations) with excellent resistance to each herbicide, the authors do not explain the motivation for exploring other indirect mechanisms of resistance, other than an improved understanding of plant physiology. Is there a potential disadvantage of CRISPR engineering that means that this approach may become unsuitable in future? Please address this point.
Other topics that I would have liked to have seen were (1) a projection of the likely future of herbicide resistance technology for weed control given that the use of herbicides (though necessary at present) is fundamentally undesirable in terms of their ecological effects. For example, the use of RNAi approaches to specific weed control (similar questions can be asked about pest resistance of course. (2) What other upcoming candidate herbicides are being explored (other than the three classes discussed) and what might their prospects be? (3) What impacts do current approaches such as transgenics have on plant yield and viability? (4) How do herbicide use, pest control, fertiliser use and the soil microbiome integrate in real crop scenarios and what is known of the effect of herbicides on these other aspects of cropping? A detailed discussion is not needed but some pointers to the wider literature would be appreciated.
Response 1: Non-target-site resistance (NTSR) mechanisms generally refer to all resistance mechanisms other than target-site resistance mechanisms. Their core characteristic is their ability to reduce the herbicide dose that reaches the target site through multiple pathways, enabling them to efficiently confer resistance to a variety of herbicides. However, research on NTSR mechanisms currently lags behind that on target resistance: the major regulatory genes have not yet been identified, nor have effective targets been identified for the creation of resistant germplasm through CRISPR gene-editing technology. Nevertheless, we anticipate that continued research will eventually reveal the major regulatory genes that govern NTSR mechanisms, clarify key functional targets, and make use of CRISPR gene-editing technology to develop new rapeseed germplasm with highly efficient herbicide resistance.
The off-target effects associated with CRISPR technology are a significant concern. The CRISPR system recognizes and cleaves target DNA sequences through the guide RNA (sgRNA) and Cas protein. However, the sgRNA may partially complement and bind to non-target genomic sequences, leading to unintended cleavage by the Cas protein and unexpected gene mutations. In crop breeding, such off-target effects can result in genomic instability, undesirable phenotypes (e.g., growth retardation and decreased disease resistance), and potential implications for crop safety. In addition, CRISPR editing efficiency is influenced by various factors, including the target species, gene locus, cell type, and editing approach (e.g., knockout, insertion, single-base substitution), exhibiting notable “preferences.” Although these limitations of CRISPR technology are not insurmountable, they do constrain its future application scenarios. Nonetheless, with ongoing technological advances such as the development of high-fidelity Cas proteins and the enhancement of off-target detection methods, these challenges are gradually being addressed. In the future, rather than being completely replaced, CRISPR is more likely to be integrated with other technologies, such as synthetic biology and multi-omics analyses, to play a pivotal role in precise crop breeding.
Response 2: We are very glad that you have raised questions about these other topics. Your ideas are excellent, and we have also considered some of these issues.
(1) Herbicide application will always effect the ecosystem to some degree, but it currently remains necessary for weed control in the field. To maintain ecological balance, we also hope that better measures can be implemented for field weed control in the future. As you mentioned, RNAi technology can be used to produce a reagent that, when sprayed on weeds, can generate double-stranded RNA that silences gene expression and achieves a weeding effect. This technology offers an eco-friendly alternative that could mitigate the ecological effects of conventional herbicides. Your proposal has expanded our research horizons. With widespread adoption, this technology could substantially benefit modern agriculture. In addition, we note that other researchers have used a “laser + AI + robot” system in which robots autonomously perform tasks ranging from AI-based weed recognition to laser weed control.
(2) In addition to the three primary herbicide categories mentioned previously, the newly developed HPPD herbicides, benquitrione and pyraquinate, effectively target broadleaf weeds. However, there has been little research on their resistance mechanisms, and no herbicide-resistant rapeseed germplasm has yet been identified. Once resistance mechanisms are fully understood, CRISPR gene editing could be used to develop herbicide-resistant rapeseed germplasm.
(3) The development of herbicide-resistant germplasm through existing technologies has not significantly reduced crop yield and viability. For instance, the IMI-herbicide-resistant rapeseed germplasm M9 exhibits normal growth and development during field cultivation, with a yield increase of 322–394 kg/ha. Jietian is a Chinese name meaning “field free of weeds.” Jietian rice varieties carry the Trp548Met mutation of the ALS gene, which confers resistance to imidazolinone herbicides. Field studies have indicated that this mutation has no detectable effect on rice growth and yield.
(4) Herbicide application, pest control, fertilizer use, and soil microbial communities interact dynamically during crop production. Herbicides can directly alter the microbial community structure by inhibiting beneficial microorganisms like rhizobia and mycorrhizal fungi, whereas other microbes decompose herbicides, mitigating their residual toxicity. These interactions influence both herbicide efficacy and soil micro-ecology. In pest control, chemical insecticides may inadvertently suppress soil antagonistic bacteria such as Bacillus, increasing disease risks. Excessive chemical fertilizers can lead to soil acidification or nutrient imbalances, hindering acid-sensitive microorganisms, whereas organic fertilizers can increase microbial diversity by supplying carbon sources. Microorganisms, in turn, enhance fertilizer transformation through nitrogen fixation and phosphorus solubilization. Cross-interactions also occur; for instance, post-herbicide weed removal necessitates organic fertilizers to counteract microbial inhibition and maintain nutrient supply. Prolonged reliance on a single approach may degrade microbial communities and induce soil fatigue.
Herbicides can have multifaceted effects on crop production. At the crop level, they may produce direct phytotoxic effects, such as leaf malformation and growth stagnation, by disrupting hormone balance or enzyme activity or by inhibiting root development and reducing nutrient absorption capacity. In the soil environment, herbicides can suppress the microorganisms responsible for organic matter decomposition, delaying straw breakdown, and may alter soil pH or the bioavailability of heavy metals. Ecologically, broad-spectrum herbicides can reduce farmland biodiversity and affect pollinating insects. Herbicides with prolonged residual activity may also cause phytotoxicity in subsequent sensitive crops. Prolonged reliance on a single herbicide can accelerate the evolution of weed resistance, posing a threat to sustainable agricultural production.
Because these topics are not the focus of this review, the relevant content is not highlighted in the article. Nevertheless, it is still a great honor to communicate with you about herbicide-related content.
Round 2
Reviewer 3 Report
Comments and Suggestions for Authors
None